# Measuring the shape of the biodiversity-disease relationship across systems reveals new findings and key gaps

Fletcher W. Halliday[1]* & Jason R. Rohr[2]

Diverse host communities commonly inhibit the spread of parasites at small scales. However, the generality of this effect remains controversial. Here, we present the analysis of 205 biodiversity–disease relationships on 67 parasite species to test whether biodiversity–disease relationships are generally nonlinear, moderated by spatial scale, and sensitive to under-representation in the literature. Our analysis of the published literature reveals that biodiversity–disease relationships are generally hump-shaped (i.e., nonlinear) and biodiversity generally inhibits disease at local scales, but this effect weakens as spatial scale increases. Spatial scale is, however, related to study design and parasite type, highlighting the need for additional multiscale research. Few studies are unrepresentative of communities at low diversity, but missing data at low diversity from field studies could result in under-reporting of amplification effects. Experiments appear to underrepresent high-diversity communities, which could result in underreporting of dilution effects. Despite context dependence, biodiversity loss at local scales appears to increase disease, suggesting that at local scales, biodiversity loss could negatively impact human and wildlife populations.

---

[1] Department of Evolutionary Biology and Environmental Studies, University of Zurich, Winterthurerstrasse 190, 8057 Zurich, Switzerland. [2] Department of Biological Sciences, Eck Institute of Global Health, Environmental Change Initiative, 180 Galvin Life Science Center, University of Notre Dame, 46556 Notre Dame, IN, USA. *email: fletcher.w.halliday@gmail.com

Understanding whether there is a general relationship between biodiversity and disease risk is critical for projecting and reducing the impacts of future disease outbreaks[1–4]. If increasing biodiversity generally reduces disease, a phenomenon coined the dilution effect, then biodiversity loss could have negative consequences for human and wildlife populations[1,5]. However, if biodiversity–disease relationships are idiosyncratic or context dependent, then biodiversity loss could have no effect on, or, in the case of an amplification effect, even reduce the risk of disease to wildlife and humans[6,7]. Such context dependence in the biodiversity–disease relationship has become a major concern among disease ecologists[3,4,8]. Consequently, numerous empirical studies and reviews of biodiversity–disease relationships have sought to define the specific conditions under which dilution or amplification are likely to take place[9–14,15]. Yet despite these studies, the degree of context dependence in biodiversity–disease relationships remains unknown. Thus, the value of biodiversity as a buffer against disease risk has been called into question[8,16,17].

Context dependence in the biodiversity–disease relationship can arise when the shape of the biodiversity–disease relationship is nonlinear. By definition, parasites require hosts for food and habitat. Thus, all else being equal, an increase in host biodiversity from zero hosts must initially increase the risk of disease[18,19]. However, if parasites are selected to infect the most abundant and widespread hosts or there are trade-offs between defending against parasites and host growth, reproduction, and dispersal, then communities might assemble in a manner where the first species added to communities are generally competent, disease-amplifying hosts and later additions might be rarer, diluting hosts[10]. If so, the initial increase in disease risk when moving from zero hosts to a few might reverse at higher diversity levels (Fig. 1a), and the skew of the biodiversity–disease relationship might affect the predominance of amplification or dilution. When biodiversity–disease relationships are left-skewed or asymptotic, amplification effects should predominate, because most increases in biodiversity will be associated with increased parasite abundance[18] (Fig. 1a). Alternatively, when biodiversity–disease relationships are right-skewed, dilution should predominate[18] (Fig. 1b). Nevertheless, understanding the shape of nonlinear biodiversity–disease relationships remains a major research gap[8,10,19,20].

Where communities fall on nonlinear biodiversity–disease curves is also important. If changes in biodiversity all occur to the right or left of the peak of unimodal diversity-disease curves, then dilution or amplification, respectively, will be most common, regardless of the direction of the skew of that relationship (Fig. 1b). This might create biases for both observational and manipulative studies. Observational studies might not capture low host diversity levels if they are rare in nature, which could lead researchers to spuriously conclude that there is a linear dilution effect, even though amplification might occur at non-sampled low levels of host diversity[8,20] (Fig. 1b). In contrast, manipulative studies might include mostly low levels of biodiversity because of the logistical challenges of collecting sufficient numbers of many host species for an experiment; this could potentially bias results towards amplification. Because the minimum diversity in any system is bounded at zero, the former bias

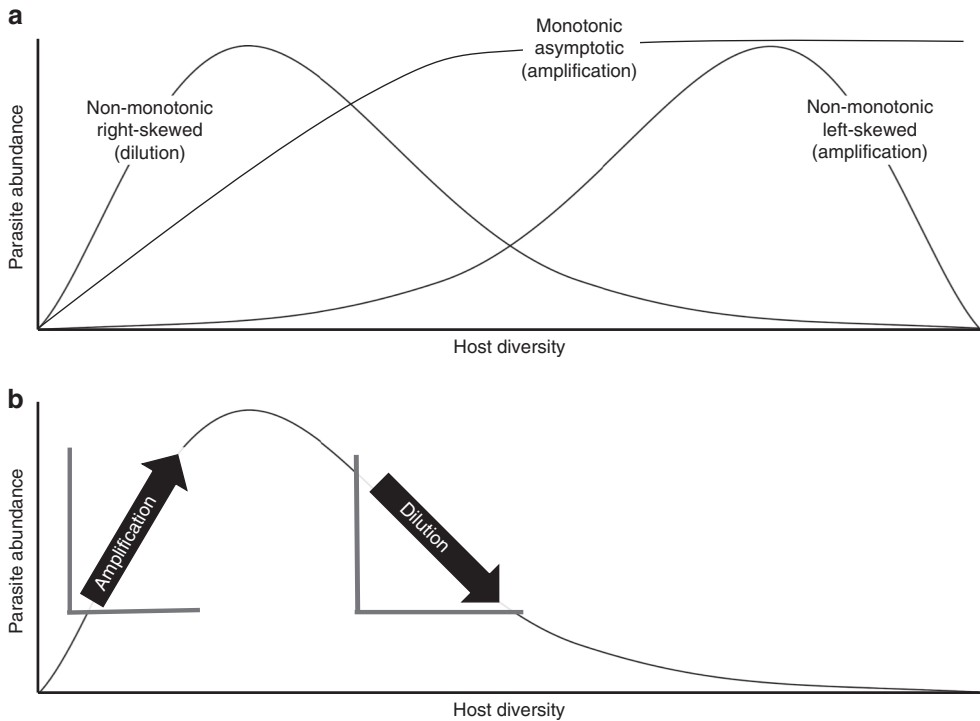

**Fig. 1** Hypothetical relationships between biodiversity and disease risk. **a** A non-monotonic right-skewed distribution suggests that dilution might occur more frequently, but less intensely than amplification because the relationship is moderately negative over a greater portion of the biodiversity gradient than it is strongly positive. A non-monotonic left-skewed distribution suggests that amplification might occur more frequently but less intensely than dilution, because the relationship is moderately positive over a greater portion of the biodiversity gradient than it is strongly negative. A monotonic and asymptotic distribution suggests that amplification becomes increasingly moderate with biodiversity. **b** In addition to the shape of biodiversity–disease relationships, the location on the curve where biodiversity levels are observed will also affect the likelihood and intensity of dilution and amplification. For example, in a right-skewed biodiversity–disease relationship, collecting measurements at biodiversity beyond the peak of parasite abundance could lead researchers to conclude that there is was a linear dilution effect, whereas measurements before the peak of parasite abundance would lead researchers to conclude that there was a linear amplification effect

can be easily detected by simply quantifying the minimum diversity level in observational studies. However, because maximum diversity is unbounded and underreported, the latter bias of experimental studies might be more difficult to assess.

Context dependence in the biodiversity–disease relationship may also arise when the direction of the biodiversity–disease relationship depends on the spatial scale of observation[2,8,21]. Local processes influence the abundance of species at relatively small spatial scales, whereas regional processes influence the distributions of species across large spatial extents[22]. Relying on this well-characterized ecological phenomenon, it has been proposed that biodiversity–disease relationships should be strongest at local scales, where biotic interactions are most likely to occur, and should weaken or could even reverse at larger scales, where individual studies encompass a greater diversity of habitat types and abiotic factors like climate may cause the distributions of hosts and parasites to covary[10,23]. In other words, at small spatial scales, increasing biodiversity might cause a reduction in parasite abundance, resulting in an observed dilution effect, whereas at larger spatial scales, regional processes might cause host biodiversity and disease risk to positively covary, offsetting the dilution effect or even resulting in an apparent amplification effect. Moreover, whether hosts can dilute disease might be more observable at small scales where encounter reduction can occur, whereas the amplifying effect of hosts might only be observable at larger temporal and spatial scales[24]. Even though theory indicates that spatial scale can moderate biodiversity–disease relationships, and biodiversity–disease studies have occurred from global to local scales[25–27], few studies have been conducted across multiple spatial scales. Thus, the degree to which biodiversity–disease relationships are moderated by spatial scale remains largely untested (but see ref.[24]).

By quantifying the shape and direction of 205 published biodiversity–disease relationships, this study aims to test three contingencies to biodiversity–disease relationships. Specifically, we test whether previously published biodiversity–disease relationships are generally (a) nonlinear, (b) moderated by spatial scale, and (c) sensitive to underrepresentation in the literature of extremely low and high diversity. Our results indicate that, among published data, biodiversity–disease relationships are generally nonlinear, that dilution most commonly occurs at small (i.e., local) scales and amplification most commonly occurs at large (i.e., regional) scales, that few studies are unrepresentative of communities at low diversity, but that missing data at low diversity in field studies could potentially result in the underreporting of amplification effects, and that experimental studies might be unrepresentative of communities at the highest diversity levels, which could potentially result in underreporting of dilution effects.

## Results and discussion

**Nonlinearity in the biodiversity–disease relationship.** First, we tested whether the published relationship between biodiversity and disease was linear or nonlinear by comparing intercept-only, linear, second-order, and third-order polynomial regression models for all biodiversity–disease relationships, selecting the best-fitting model using Akaike's information criterion (AIC) (Supplementary Data 1). Importantly, these models were only based on the data presented in each study and thus did not constrain the biodiversity–disease relationship to the origin (see 'Do missing data at low and high-diversity bias studies to report dilution effects?' section below). Out of the 205 studies that included more than three levels of biodiversity, 67% were best fit by a linear, second-order, or third-order polynomial model (i.e., exhibited a relationship between biodiversity and disease). Of these studies,

biodiversity–disease relationships were most commonly nonlinear, as predicted. More specifically, 61% exhibited nonlinear relationships (either second- or third-order polynomial), whereas 6% exhibited a linear, positive biodiversity–disease relationship (e.g., linear amplification effect), and 33% exhibited a linear, negative biodiversity–disease relationship (e.g., linear dilution effect). Whether the best-fitting model was linear, nonlinear, or intercept-only did not depend on the number of unique values of host diversity in a study (Supplementary Data 1). These effects were also qualitatively similar using two goodness-of-fit statistics: the adjusted $R^2$ (80% of studies were best fit by a linear, second-order, or third-order polynomial model; of these, 71% exhibited nonlinear relationships, either second- or third-order polynomial functions) and the more conservative, Bayesian information criterion (BIC) for model selection (49% were best fit by a linear, second-order, or third-order polynomial model; of these, 53% exhibited nonlinear relationships; Supplementary Note 1; Supplementary Data 1).

Although comparing regression models identified many nonlinear biodiversity–disease relationships, this approach is constrained by the functional form of each regression model. In other words, we are only able to detect nonlinear relationships where those relationships were best fit by second- or third-order polynomials. To relax this constraint, we used Spearman rank correlation tests (not constrained to pass through the origin), which make no assumption about the underlying distribution of the data nor the linearity of the relationship between variables, and are therefore not constrained by the functional form of the biodiversity–disease relationship. We quantified whether each biodiversity–disease relationship was monotonic and positive (disease increases, but may level off, as diversity increases), monotonic and negative (disease decreases but may level off as diversity increases), or non-monotonic (disease increases with diversity at low levels, but eventually decreases at high enough diversity; Fig. 1a). The estimated Spearman rank correlation coefficient ($\rho$) approaches one for monotonic, positive relationships, and approaches negative one for monotonic, negative relationships. We therefore used $\rho$ to define monotonic amplification ($\rho > 0$, Spearman $p < 0.05$), monotonic dilution ($\rho < 0$, Spearman $p < 0.05$), and non-significant or non-monotonic relationships (Spearman $p > 0.05$). Consistent with the previous analysis, 10% of the 205 relationships exhibited monotonic amplification effects, 35% exhibited monotonic dilution effects, and 55% exhibited non-significant or non-monotonic relationships.

Given that nonlinear and non-monotonic biodiversity–disease relationships are most common and that amplification effects might predominate when these relationships are left-skewed or asymptotic, whereas dilution might predominate when they are right-skewed[18], we next assessed the skew of each biodiversity–disease relationship. To do so, we fit a smoothing spline to each published biodiversity–disease relationship, that was not constrained to pass through the origin, and then calculated Pearson's skewness from the shape of the estimated curve, excluding studies where there was no relationship (i.e., where the slope of the curve was not significantly different from zero; $n = 29$), because Pearson's skewness cannot be estimated from a curve with a slope of zero. As expected, Pearson's skewness and Spearman rank correlation were in agreement when studies exhibited monotonic biodiversity–disease relationships. Specifically, studies exhibiting monotonic dilution effects were significantly right-skewed ($t$ test $p < 0.001$), and studies exhibiting monotonic amplification effects were significantly left-skewed ($t$ test $p < 0.001$; Fig. 2). Studies exhibiting non-significant or non-monotonic relationships based on Spearman rank correlation were not significantly skewed ($t$ test $p = 0.80$), indicating that

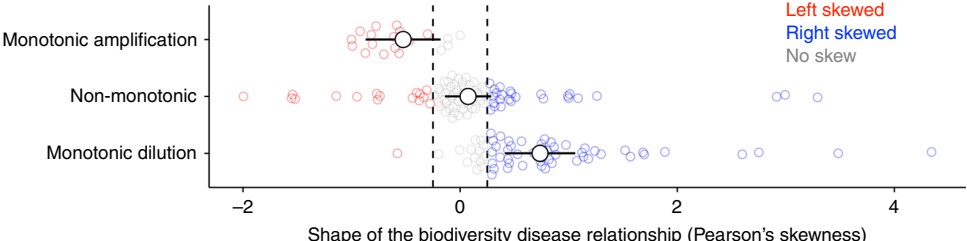

**Fig. 2** Results of the analysis comparing Spearman rank correlation to Pearson's skewness. Points are model-estimated means and error bars are 95% confidence intervals. The colored points show the distribution of the raw data. Left-skewed relationships (Pearson's skewness < 0.25) are shown in red, right-skewed relationships (Pearson's skewness > 0.25) are shown in blue, and non-skewed relationships are shown in gray. Spearman rank correlation was strongly associated with Pearson's skewness: monotonic amplification effects ($\rho > 0$, Spearman $p < 0.05$) tended to be left-skewed, monotonic dilution effects ($\rho < 0$, Spearman $p < 0.05$) were right skewed, and non-monotonic relationships were not significantly skewed. Source data are provided as a Source Data file

non-monotonic biodiversity–disease relationships, were on average, equally likely to show amplification or dilution. These results were qualitatively similar for the analysis comparing intercept-only, linear, second-order, and third-order polynomial regression models (Supplementary Figures 1, 2).

These results support the hypothesis that the shape of biodiversity–disease relationships might be nonlinear[8,18]. Thus, where an individual system falls along a biodiversity gradient might influence whether that system experiences amplification or dilution. Similarly, predicting how disturbances will alter disease may depend on where individual systems fall along a biodiversity gradient.

**Effects of spatial scale**. Second, we tested whether the shape and direction of the biodiversity–disease relationship was moderated by spatial scale. Spatial scale can be decomposed into spatial grain, which represents the area over which a single replicate measure of biodiversity and disease are collected, and spatial extent, which represents the total area over which a study is conducted, including all measures of biodiversity and disease for a given study[28]. Although spatial scale can be quantified in absolute terms (e.g., km²), comparing spatial scale among studies can be problematic, particularly if spatial scale is confounded with host biomass. For example, a study of bacteria could be carried out in a test-tube or in an ocean, but a study of whales could never be conducted in a test-tube. We expected this missing-cells design to be more problematic for studies of small spatial grain, which might include a single population of a small-bodied host organism, than for studies of small spatial extents, which must always include multiple replicate host communities, by design. We therefore report the results of unstandardized spatial extent, noting that the results were qualitatively similar for spatial extent standardized by host biomass and for spatial grain standardized by host biomass (Supplementary Note 2).

In the published literature, spatial extent tended to be correlated with the metrics used to estimate diversity and disease as well as several characteristics of individual study systems, potentially obscuring the effect of spatial scale on the shape of the biodiversity–disease relationship (Fig. 3). Specifically, biodiversity–disease relationships at the largest spatial extents were dominated by observational studies of human diseases (Fig. 3c, Fig. 3f). These results highlight a pressing need for comprehensive studies of biodiversity and disease conducted across spatial scales. Despite the lack of measurements across all endpoints and spatial scales, host richness and parasite prevalence were reported across more than five orders of magnitude in spatial extent, allowing studies to be compared across systems (Fig. 3a, Fig. 3b). However, the following results should still be

interpreted with caution, as extreme values of spatial extent may be confounded with other characteristics of study systems.

Spearman's $\rho$, which measures the monotonicity and direction of association between biodiversity and disease, was positively associated with spatial extent (Type III ANOVA $p = 0.002$; marginal $R^2 = 0.16$; Fig. 4a), with monotonic dilution effects most commonly occurring at small to intermediate spatial scales and monotonic amplification effects most commonly occurring at the largest spatial scale. Incorporating the shape of non-monotonic relationships did not alter this result; Pearson's skewness was significantly associated with spatial extent (Type III ANOVA $p < 0.001$; marginal $R^2 = 0.15$; Fig. 4b), with right-skewed relationships (indicating more dilution) occurring at small to intermediate spatial scales and left-skewed relationships (indicating more amplification) occurring at large spatial scales. This effect was not moderated by how host diversity and disease were measured (Supplementary Table 1).

Right-skewed and negative-monotonic relationships generally occurred within an ecosystem, at spatial extents < 100 km² (roughly the size of a small city), whereas left-skewed and positive-monotonic relationships generally occurred across ecosystems, in studies occupying > 1,000,000 km² (roughly the size of France and Spain combined). However, the analysis of spatial grain did not indicate significant amplification at any spatial scale (Supplementary Note 2), and so results regarding amplification at large scales should be interpreted with caution. We did find significant amplification at the largest spatial extents, indicating that the overall disease burden in one ecosystem can be higher than another because its native biodiversity is higher, but if this ecosystem has its biodiversity lowered, disease could still worsen.

This dependence of biodiversity–disease relationships on spatial scale may be an indicator of a more general mechanism of disease amplification. Notably, comparison of biodiversity–disease relationships within an ecosystem often include many of the same host species, e.g.,[29] whereas comparisons of biodiversity across ecosystems tend to include distinct sets of host species, e.g.,[30]. Thus, measuring the degree to which host–species turnover ($\beta$ diversity) drives biodiversity–disease relationships could help clarify why amplification was more commonly observed at large spatial extents, and could possibly help predict when amplification will be more common, in general.

These results reveal an association between the shape of published biodiversity–disease relationships and the spatial scale of observations, supporting the hypothesis that biodiversity–disease relationships are scale-dependent[2,10]. Importantly, however, not every small-scale study exhibited dilution, nor did every large-scale study exhibit amplification. As an example, at small spatial scales, biodiversity can amplify disease via a sampling effect if species are added randomly with respect to host competence and transmission

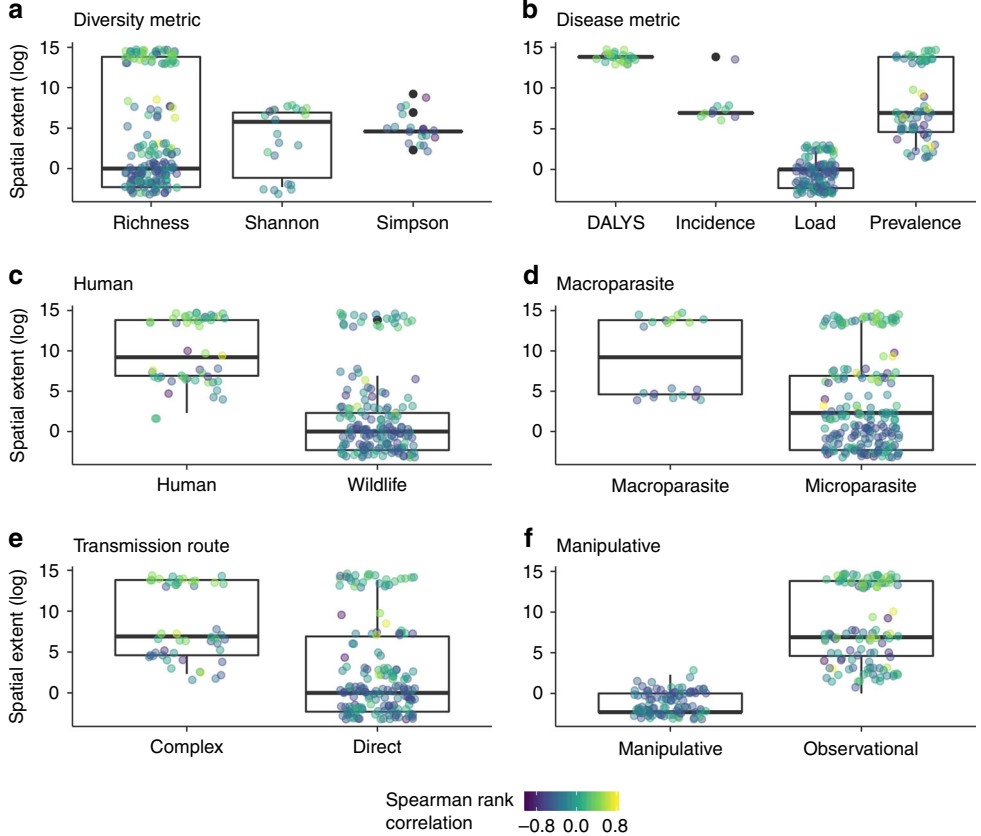

**Fig. 3** Relationship between spatial extent and ecological factors. **a** host diversity metric, **b** disease metric, **c** parasites that infect humans vs. wildlife, **d** macro- vs. microparasites, **e** parasites with complex vs. direct lifecycles, and **f** observational vs. manipulative studies. Each point represents an individual study, colored by the Spearman rank correlation coefficient for the study, with numbers below zero (purple and dark-blue) indicating monotonic dilution and numbers above zero (light-green and yellow) indicating monotonic amplification effects. The box shows the first and third quartiles, the middle line shows the median, and the whiskers extend from the box to the largest and smallest values, no more than 1.5 × the interquartile range. Source data are provided as a Source Data file

is frequency-dependent[9], and in a global survey of island nations, leptospirosis incidence strongly declined with increasing mammal species richness[31].

We performed a sensitivity analysis to test whether the observed spatial moderation of biodiversity–disease relationships depended on any of the three studies that were conducted at the largest spatial scale (Supplementary Note 2). One study, a global survey of human disease burden[30], strongly influenced the results of our analysis. Omitting this study from the analysis of spatial scale did not qualitatively change the significant effect of spatial extent on the skewness of the biodiversity–disease relationship (Type III ANOVA $p = 0.006$), but did eliminate the significant effect of spatial extent on the monotonicity and direction of the relationship (Type III ANOVA $p = 0.27$). Importantly, despite the sensitivity of these analyses to studies at the largest spatial scale, the effect of spatial scale on the shape and direction of the biodiversity–disease relationship showed a similar trend even when we excluded all three influential data sets at the largest spatial scale from our analysis. Specifically, after omitting the three studies conducted at the largest spatial extent, we still detected significantly monotonic negative and right-skewed biodiversity relationships only at spatial extents below < 100 km² (roughly the size of a small city).

At a given spatial scale, ecological factors including characteristics of host and parasite species can influence whether dilution or amplification are observed[32–34]. Accounting for characteristics of host species, genotypes, or life-history

stages that are driving biodiversity–disease relationships both within and across spatial scales will therefore be critical for future studies aimed at developing a more mechanistic understanding of biodiversity–disease relationships across scales. We tested whether several ecological factors could explain variation in the effect of spatial scale on the shape of biodiversity–disease relationships. Specifically, we tested whether the effect of spatial scale on biodiversity–disease relationships differed between (i) parasites that infect humans vs. wildlife, (ii) macro- vs. microparasites, (iii) parasites with complex vs. direct lifecycles, and (iv) observational vs. manipulative studies. We found no evidence that the effect of spatial scale on biodiversity–disease relationships was moderated by any of these factors (Table 1). Thus, the effect of spatial scale on biodiversity–disease relationships was generally robust across all ecological contexts examined.

Despite the generally robust effect of spatial scale on the shape of biodiversity–disease relationships, we still encourage caution in interpreting these results, as there was multicollinearity in these analyses. Specifically, observational studies and studies of human pathogens both tended to occur at larger spatial scales than manipulative studies and studies of wildlife pathogens (Fig. 3). This collinearity highlights an important limitation in the study of biodiversity–disease relationships: our understanding of the relationship between biodiversity and disease is limited by research priorities, approaches, and study systems, which can vary among individual research groups[2,10]. Consequently, we

cannot rule out the possibility that these results could change if future studies filled these research gaps, allowing tests of these

context dependencies to be less collinear. Furthermore, the scale associated with data used does not necessarily mean that this is the scale of the transmission cycle, and ignoring the spatial scale of transmission can lead to spurious conclusions. For example, depending on the scale of transmission, study scale can determine whether wildlife loss protects against or promotes tick-borne disease[24]. It is therefore possible that some studies conducted at small spatial scales did not capture the entire parasite transmission cycle compared with studies conducted at larger spatial scales. Ideally, an analysis of spatial scale would therefore include transmission scale as a standardizing variable. However, because the scale of transmission is unreported or unknown for most pathogens, we instead used host biomass, because transmission often scales with host biomass[35,36]. Our results were robust to standardization by host biomass (Supplementary Figure 3).

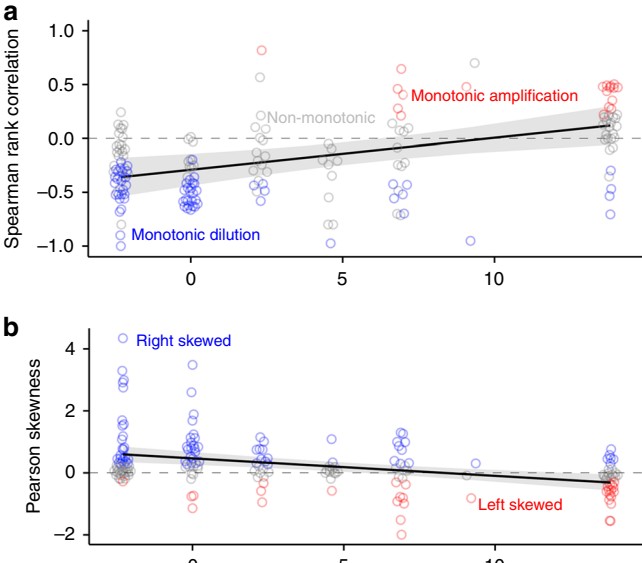

**Fig. 4** Results of the analyses relating spatial scale to the shape of the biodiversity–disease relationship. Points represent each published biodiversity–disease relationship, colored by their estimated shape (red = monotonic amplification in **a** and left-skewed in **b**; blue = monotonic dilution in **a** and right-skewed in **b**; gray = non-significant or non-monotonic in **a** non-skewed in **b**). Solid lines indicate the estimated fit of a multilevel random effects model, and gray ribbons indicate the 95% confidence intervals. Spatial scale moderates the relationship between biodiversity and disease: **a** Spearman rank correlation between biodiversity and disease was positively associated with spatial extent, and **b** Pearson's skewness was negatively associated with spatial extent. Source data are provided as a Source Data file

**Underrepresentation in the literature.** Finally, we tested the hypothesis that underrepresentation in the literature of extreme environmental scenarios with either high or low biodiversity communities in experimental and observational studies might bias studies to more commonly report amplification and dilution effects, respectively. Experimental studies had a lower mean maximum diversity level than observational studies (experimental mean ± sd: 25 ± 10, observational mean ± sd: 48 ± 74). Thus, it appears that experimental studies are under representing communities at the highest diversity, which could bias experimental studies towards amplification effects. This result could emerge from two key differences between experiments and observational studies. First, experimentally manipulating many species is logistically challenging at high richness, potentially biasing experimental studies to include fewer total species than observational studies of equivalent size. Second, the number of species in an area is highly sensitive to the area surveyed[37], and observational studies were, on average, four orders of magnitude larger than manipulative experiments (Supplementary Figure 4). Focusing on studies of comparable extent (1–10 km$^2$) eliminated the difference in mean maximum diversity between experiments (29 ± 10) and observational studies (22.0 ± 10), supporting this second mechanism.

We also examined the lowest diversity levels to assess whether there was underrepresentation in the literature of environmental scenarios with low diversity. Experimental studies had lower mean minimum diversity than observational studies (experimental mean ± sd: 1.2 ± 0.7; observational mean ± sd: 4.9 ± 8.0), which could bias observational studies towards dilution effects. However, 82% of the 205 studies included a measurement of effective species richness of two or lower. Consequently, as effective species richness represents the effective number of species in a community (which is bounded at zero), most studies ($n = 168$) were not missing substantial data at low host diversity. This result indicates that the potential for underrepresentation in the literature at extremely low diversity to bias the estimated relationship between biodiversity and disease is quite low.

Even though most studies were not under representative of communities at low host diversity, we still performed an additional test of the hypothesis that underrepresentation in the literature might bias studies to more commonly report dilution effects. Here, we again quantified the skew of each biodiversity–disease relationship, this time constraining each curve to pass through the origin, because if there are no hosts there cannot be any parasites. Constraining each curve to pass through the origin should reduce the estimated skew in all studies, particularly studies that found monotonic dilution effects. As predicted, constraining the curves to the origin significantly changed the shape of the average biodiversity–disease relationship

**Table 1 Models of ecological factors moderating the effect of spatial scale on biodiversity–disease relationships**

|  | DF | F value | p value |
|---|---|---|---|
| *Spearman correlation coefficient* |  |  |  |
| Spatial extent | 32.9 | 1.206 | 0.28 |
| Human | 39.5 | 0.116 | 0.74 |
| Route | 39.2 | 1.719 | 0.19 |
| Macroparasite | 29.0 | 1.055 | 0.31 |
| Manipulative | 38.4 | 0.262 | 0.61 |
| Spatial extent × human | 34.4 | 0.073 | 0.79 |
| Spatial extent × route | 38.2 | 0.269 | 0.61 |
| Spatial extent × macroparasite | 30.5 | 0.576 | 0.45 |
| Spatial extent × manipulative | 33.9 | 0.000 | 0.99 |
| *Pearson's skewness* |  |  |  |
| Spatial extent | 6.9 | 0.177 | 0.69 |
| Human | 47.4 | 0.100 | 0.75 |
| Route | 39.2 | 0.489 | 0.48 |
| Macroparasite | 9.4 | 0.140 | 0.71 |
| Manipulative | 12.7 | 0.004 | 0.94 |
| Spatial extent × human | 26.7 | 0.076 | 0.78 |
| Spatial extent × route | 52.6 | 0.374 | 0.54 |
| Spatial extent × macroparasite | 18.4 | 0.285 | 0.59 |
| Spatial extent × manipulative | 5.6 | 0.001 | 0.97 |

Type III analysis of variance table with Satterthwaite's method
DF, denominator degrees of freedom

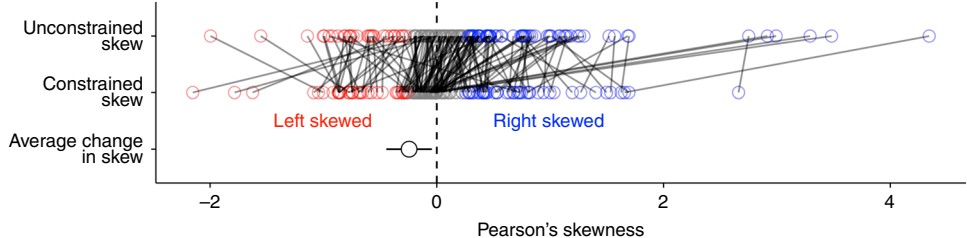

**Fig. 5** Results of constraining biodiversity–disease relationships to pass through the origin. The top two rows show Pearson's skewness for unconstrained curves, and curves that were constrained to pass through the origin, with each study connected by a solid line. Left-skewed relationships (Pearson's skewness < 0.25) are shown in red, right-skewed relationships (Pearson's skewness > 0.25) are shown in blue, and non-skewed relationships are shown in gray. The bottom row shows the model-estimated effect of constraining the curves to pass through the origin, with the point indicating the model-estimated mean, and error bars showing the 95% confidence interval. On average, constraining curves to pass through the origin results in a more left-skewed relationship between biodiversity and disease. Source data are provided as a Source Data file

($t$ test $p < 0.001$), reducing the estimated frequency of dilution effects and increasing the estimated frequency of amplification effects (Fig. 5). This result indicates that underrepresentation of communities at low host diversity may bias some studies to underreport amplification effects. However, even though constraining curves to fit through the origin shifted the estimated skew of most studies, on average, the constrained curves were not significantly left-skewed ($t$ test $p = 0.55$; Supplementary Figure 5). Furthermore, spatial scale still significantly moderated the sign of the constrained curves, with dilution more common at small scales and amplification more common at large scales (Type III ANOVA $p < 0.001$; Supplementary Figure 4), and this effect was still robust to ecological characteristics of individual study systems (Supplementary Table 2).

These results indicate that scale-dependence of published biodiversity–disease relationships is robust to underrepresentation of communities at low diversity levels. The robustness of this scale-dependence may be a product of the underlying shape of biodiversity–disease relationships. Studies that found monotonic amplification effects were unlikely to be altered by missing data at low diversity (Fig. 1a). Conversely, studies that found monotonic dilution had higher potential to be altered by missing data at low diversity. However, 66 of the 72 studies showing monotonic dilution effects included an effective species richness of two or lower. Thus, regardless of the shape of the relationship between the origin and the point of peak parasite abundance, the area in which amplification could occur was generally quite small. We therefore conclude that although the biodiversity–disease relationship can take on many forms, and its form may depend on a nonlinearity that is driven by parasite extinction at low host diversity, such nonlinearities are unlikely to alter a general and common phenomenon: dilution effects are most commonly observed at local scales and the effect weakens and may even reverse as spatial scale increases.

Together, these results indicate that the scale-dependence of biodiversity–disease relationships might be robust to underrepresentation of communities at low diversity levels and to ecological characteristics of individual studies. However, we are unable to test whether underrepresentation of communities at high diversity might bias experimental studies to more commonly observe amplification. This bias of experimental studies is difficult to assess because the maximum biodiversity in a host community is unbounded and underreported, and in most systems, the distribution of natural levels of host diversity is unreported. Furthermore, there is still considerable debate surrounding the scale, location, and context of biodiversity loss[38]. Thus, the applied significance of missing biodiversity data in experimental studies remains unresolved.

The results of this analysis are limited by the availability of previously published data, which may be biased to report significant effects (i.e., a "file-drawer problem") or to select systems to study that are likely to show dilution (i.e., to show "system selection bias")[2,39]. However, a previous study conducted in 2015 found no evidence of a file-drawer problem for parasites of humans[40]. Furthermore, by re-analyzing previously published data, our study is somewhat less sensitive to the file-drawer problem than typical meta-analyses, which rely on previously published statistical tests. Nevertheless, we cannot rule out the possibility for an overabundance of studies finding significant dilution or amplification relative to the frequency of dilution or amplification in nature.

A major limitation to this analysis and a major research gap in the study of biodiversity–disease relationships is the lack of empirical studies conducted across spatial scales and ecological conditions using the same methodologies. Specifically, although we found a significant, positive relationship between spatial extent and the direction of the biodiversity–disease relationship, the spatial extent of a study tended to be non-independent of study design and parasite type. Furthermore, our models only estimated significant amplification when the spatial extent of a study was >1,000,000 km², and only three studies have measured biodiversity–disease relationships at this scale, one of which contributed nearly every estimate of amplification and used a unique and human-specific disease metric (DALYs, Fig. 3)[30]. Our analysis indicates that the effect of spatial scale on the shape of the biodiversity–disease relationship is robust to the omission of this study. However, even though the relationship between spatial extent and the shape of biodiversity–disease relationships was generally robust in our analysis, we cannot rule out the possibility that additional multiscale studies, particularly at the largest spatial extents, could change these results. Consequently, the existing literature may be insufficient to predict how biodiversity loss at the largest spatial scales (e.g., via mass extinctions) will alter disease risk. We are hopeful that as large-scale replicated studies, such as the Nutrient Network[41] and National Environmental Observatory Network[42], become more widespread, the quality of data at the largest spatial scales will improve.

Understanding how biodiversity alters infectious diseases remains a critical frontier in disease ecology[10,11,43] as human activities continue to alter global biodiversity[44,45], and disease outbreaks continue to increase[46–48]. This study provides quantitative evidence that, among published studies, the relationship between biodiversity and disease is nonlinear and scale-dependent. This general pattern indicates that biodiversity loss could exacerbate disease outbreaks at the scales in which humans are most likely to encounter disease, and highlights important

## Methods

**Data compilation**. This study aimed to analyze the shape of every published relationship between host diversity and the abundance of parasites. We updated the list of studies from Civitello et al.[3] to include studies published between 2014 and 2018, by repeating their original search criteria. Specifically, we searched the Web of Science for several combinations of search terms: parasite, pathogen, diversity, richness, evenness, dilution effect, and amplification effect (the final search was conducted in June 2018). We identified additional studies by searching the literature cited sections of these articles and by searching Web of Science for all papers citing Civitello et al.[3], including those critical of the dilution effect hypothesis. We included observational and experimental studies in lab and field environments.

**Selection criteria and data collection**. We only included studies that measured parasite abundance or prevalence at more than two host diversity levels. We included studies that reported infection prevalence, mean parasite load, density of infected vectors, or percent diseased tissue, because these quantities are the most relevant metrics of disease risk for microparasites, macroparasites, vector-borne parasites, and plant parasites, respectively[3]. We did not standardize parasite abundance, as standardization would not alter the estimated shape of the biodiversity–disease relationship, and we therefore did not compare parasite abundance among studies. Host biodiversity was reported as species richness, Simpson's diversity index ($J$), or Shannon's diversity index ($H$). We standardized across these measures to facilitate comparisons across studies by transforming diversity into the effective number of species, following Jost[49]. This transformation puts species richness, Simpson's diversity index, and Shannon's diversity index on the same arithmetic scale, but does not change the underlying data that were used to calculate each metric. In experiments, estimated diversity included all taxa added by the experimenters, whereas the diversity estimate in observational studies was limited to a focal taxonomic or functional group of host species, defined in the primary study (e.g., herbaceous plants, trees, birds, or small mammals).

We extracted data from text and tables manually and from figures using WebPlotDigitizer version 4.1[50], and recorded other data relating to the biology or methodology of each study. For all studies, we recorded parasite and host taxa, type of parasite (infecting only wildlife or also infecting humans), focal host species, associated species (i.e., additional species whose presence may dilute or amplify parasite abundance, operationally defined as 'potential diluters'), the diversity (e.g., richness) in the treatments (or in the field survey), parasite functional group (macroparasite vs. microparasite), parasite lifecycle (complex vs. direct), and study design (manipulative vs. observational). Spatial extent was quantified as the area (expressed in square kilometers) over which all biodiversity estimates were compared in a given study. This measurement is distinct from spatial grain, which was quantified as the size of an individual host community. Thus, if there were multiple small plots within a large region, the extent of the study would be quantified as large, whereas grain of the study would be quantified as small. Studies rarely provided an exact value for spatial extent. Because a value for spatial extent was rarely provided, and spatial extents varied by six orders of magnitude, we estimated the extent of each survey to the nearest order of magnitude rather than attempting to assign a specific spatial extent for each study. For example, we assigned studies a value of 0.1 if the extent was <1 km$^2$, and a value of 1 if the extent was >1 km$^2$, but <10 km$^2$, etc. When a study included more than one grain size (i.e., when spatial grain varied among communities in a study), we used the average grain size as our estimate for that study.

**Assessing the shape of the biodiversity–disease relationship**. To standardize our assessment of biodiversity–disease relationships among studies, we analyzed the shape of each biodiversity–disease relationship independently using standardized methodology, and then compared the shapes analytically. Specifically, we first quantified whether each biodiversity–disease relationship was linear or nonlinear by comparing a series of regression models using the lm, AIC, and BIC functions in R version 3.5.2[51] (see the Results and Discussion subsection 'Nonlinearity in the biodiversity–disease relationship' and Supplementary Note 1). Four studies included fewer than five host diversity levels and were therefore not tested using a third-order polynomial. Next, we quantified the monotonicity and direction of each biodiversity–disease relationship using Spearman rank correlations (see the Results and Discussion subsection 'Nonlinearity in the biodiversity–disease relationship'). In brief, the Spearman rank correlation coefficient Rho ($\rho$) and its associated $p$ value were used to define monotonic amplification ($\rho > 0$, Spearman $p < 0.05$), monotonic dilution ($\rho < 0$, Spearman $p < 0.05$), and non-significant or non-monotonic relationships (Spearman $p > 0.05$). We then assessed the skew of each biodiversity–disease relationship using R package cobs[52] to fit an unconstrained spline to the biodiversity–disease relationship, limited to a maximum of four knots to prevent overfitting. This approach to fitting an unconstrained curve makes no assumptions about the underlying shape of the relationship between biodiversity and disease. We transformed the predicted curve into a frequency distribution, assigning any negative value (occurring in 19 regressions) to zero, and then calculated Pearson's skewness. A right-skewed relationship (Pearson skewness > 0.25)

indicates that most of the data falls in the area where dilution is observed, while a left-skewed relationship (Pearson skewness < −0.25) indicates the possibility for measured or unmeasured amplification effects. To assess whether missing data at low diversity could bias the estimated shape of the biodiversity–disease relationship, we constrained curves to pass through the origin and again calculated the skewness of each curve. Specifically, to fit qualitatively constrained quantile (CQ) smoothing splines[53], we added a value at the origin for each data set, corresponding to a situation in which there is no host diversity, generated a constraint matrix to force the line through the origin, and then fit the curve, limiting the maximum number of knots in the curve to three to prevent overfitting.

We omitted studies with fewer than four unique measures of host diversity for Spearman rank correlations and unconstrained splines and fewer than three unique measures of host diversity for CQ splines. Twenty-nine of the unconstrained splines ($n = 205$) and 39 of the CQ splines ($n = 217$) showed no relationship between biodiversity and disease (e.g., a fit with a slope of zero), resulting in no estimate of Pearson's skewness. This resulted in 205 estimates of $\rho$, 176 estimates of skew from unconstrained splines, and 178 estimates of skew from CQ splines.

**Data analysis**. All analyses were carried out in R version 3.5.2[51]. We constructed multilevel random effects models using the lmer function in R packages lme4[54] and lmerTest[55]. We accounted for nonindependence arising from multiple measures from the same observational units in the same year by including such non-independent surveys as random intercepts in each model. We also included parasite species as a random intercept in each model, though in some models, parasite species explained no residual variance leading to a computational singularity, and was therefore omitted from the model.

Using the model described above, we first tested whether studies exhibiting no relationship, linear amplification, linear dilution, a unimodal relationship or a third-order polynomial relationship predicted Spearman rank correlation ($n = 205$) and Pearson's skewness ($n = 176$), by performing pairwise comparisons of the model-estimated fixed effects using R package lsmeans[56]. We next verified that studies exhibiting monotonic dilution, monotonic amplification, and non-monotonic relationships (categorized using the Spearman rank correlation) predicted Pearson's skewness ($n = 176$), by again performing pairwise comparisons of the model-estimated fixed effects. Next, we tested whether the Spearman rank correlation coefficient between biodiversity and disease or Pearson's skewness were influenced by spatial extent by fitting two separate models, each with one response ($\rho$, $n = 205$; or skew, $n = 176$) and one predictor (extent).

The effect of spatial extent on biodiversity–disease relationships may depend on the size of the host organism. We therefore estimated host body size to the nearest order of magnitude using TraitBank records from the Encyclopedia of Life, standardized spatial extent by host biomass, and then tested whether spatial extent still moderated biodiversity–disease relationships using the standardized estimates of extent. The effect of spatial extent on biodiversity–disease relationships was robust to standardization by host biomass (Supplementary Figure 3), and because spatial extent is easier to interpret when unstandardized, we report the results of unstandardized spatial extent.

The effect of spatial scale on biodiversity–disease relationships may also be sensitive to the measure of spatial scale. We therefore repeated the analyses using spatial grain, standardized by host biomass, in place of spatial extent. The effect of spatial scale was robust to the metric used (Supplementary Note 2).

The effect of spatial scale on biodiversity–disease relationships may also be sensitive to data at the largest spatial scales. We therefore repeated the analyses of spatial extent and spatial grain with and without biodiversity–disease relationships quantified in studies by Wood et al.[30], Nguyen et al.[57], and Derne et al.[31] (Supplementary Note 2).

We then tested for context dependence in the spatial moderation of dilution effects. To test for context dependence, we fit the same two models, but included a two-way interaction between spatial extent and four binary factors that might explain variation in the effects of scale on the biodiversity–disease relationship: parasite functional group (macroparasite vs. microparasite), parasite lifecycle (complex vs. direct), study design (manipulative vs. observational), and parasite type (infects humans vs. infects only wildlife). To test whether the effect of spatial scale on the shape of biodiversity–disease relationships was influenced by the type of diversity metric (e.g., host richness, Shannon diversity, or Simpson's diversity), or disease metric (e.g., prevalence, severity), we fit the same models, but included a two-way interaction between spatial scale and either the diversity metric or the disease metric.

We next tested whether underrepresentation in the literature of communities at low and high diversity might bias studies to more commonly report amplification and dilution effects. We quantified the maximum and minimum diversity level of each study and compared whether the mean maximum and mean minimum diversity level differed between experiments and observational studies. Because the species-area relationship is nonlinear and sample area was highly variable across studies, we compared minimum and maximum diversity across studies qualitatively rather than quantitatively.

To quantitatively test whether missing data at low host diversity could bias studies to more commonly report dilution effects, we tested whether constraining the curves to pass through the origin altered the predicted skew. Specifically, we calculated the difference in skew between constrained and unconstrained curves

and then performed an intercept-only model on this value, where an estimate significantly lower than zero would indicate that constraining the curve favored amplification, and an estimate significantly higher than zero would indicate that constraining the curve favored dilution. Finally, we analyzed whether spatial scale moderated the shape of the biodiversity–disease relationship when curves were constrained to pass through the origin. Here, we fit a model of Pearson's skewness and spatial extent and then performed the same test of context dependence on the model that was performed before.

**Reporting summary**. Further information on research design is available in the Nature Research Reporting Summary linked to this article.

## Data availability

The data supporting the results are archived on Figshare (https://doi.org/10.6084/m9.figshare.9784226). The source data underlying Figs. 2–5 and Supplementary Figs. 1–16 are provided as a Source Data file.

## Code availability

The code supporting the results are archived on Figshare (https://doi.org/10.6084/m9.figshare.9784226).

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

## Acknowledgements
We are thankful to D. Civitello, R.W. Heckman, G. Legault, C.E. Mitchell, J. Umbanhowar, and members of the Rohr lab for helpful discussions on data analysis and interpretation. C.E. Mitchell, R. Poulin, H. Young, and S. Zhou provided raw data from published manuscripts. This research was supported by grants from the National Science Foundation (EF-1241889), National Institutes of Health (R01GM109499, R01TW010286), US Department of Agriculture (NRI 2006-01370, 2009-35102-0543), and US Environmental Protection Agency (CAREER 83518801) to J.R. Rohr.

## Author contributions
J.R.R. & F.W.H. designed the study. F.W.H. analyzed the data and wrote the first draft. Both authors contributed substantially to revising the manuscript.

## Competing interests
The authors declare no competing interests.
