## [Peer Review File · Nature Communications]

Editorial Note: This manuscript has been previously reviewed at another journal that is not operating a transparent peer review scheme. This document only contains reviewer comments and rebuttal letters for versions considered at Nature Communications .

Reviewers' Comments:

Reviewer #2:

Remarks to the Author:

In the revised version of the manuscript, the authors have sufficiently addressed my previous comments. The manuscript addresses an interesting and significant problem, and although the data set has limitations, the authors have conducted a rigorous analysis that addresses my concerns.

Reviewer #3:

Remarks to the Author:

I find the ms much improved, but I remain unconvinced that the underlying data are sufficient and comparable enough to merit the conclusion reached (both in positive findings and in negative findings)

For instance, I am not clear how the authors analysis can control for the effects of metric on response, with these inconsistent datasets. In this case it appears the most common metric in their large scale studies (DALY) is simply not used at small scales, and is fundamentally a very different metric than all other used metrics. Similarly as far as I can tell there are NO experimental studies at large scales so I don't understand how the authors can claim evidence that there is no artifact of experiment vs. observation x spatial scale. In general what I see is lack of support for any of these factors more likely to be driven by noise, and the factors that are supported, are highly likely to be driven by biases in the underlying dataset. I think there is just a lack of sufficient comparable data to answer the questions about scale

Authors conclude as the final sentence in their abstract "that biodiversity loss could have negative consequences for human and wildlife populations." This idea recurs elsewhere in their paper Yet the authors purport to show that biodiversity loss does NOT have negative consequences (e.g. via increasing disease transmission) at large scales. Given that the scale at which biodiversity is actually declining is large (e.g. review in McGill et al 2015) and not small this seems like a major misinterpretation of their findings. Indeed, if anything their data argues the opposite – which is that at the large scale biodiversity loss is thought to be most pervasive, there are unlikely to be negative consequences for human and wildlife health.

I'm afraid I can't understand/make use of much of their data files because of missing metadata that makes it hard to understand what data was used. For instance I'm not sure what diversity "levels" are in the supp file. They do not seem to correspond with the total number of species reported in the paper cited for a couple where I checked? Although since I don't have a reference list to correlate with the papers cited I can't tell for certain I am cross checking correct papers (papers are numbered in figshare file but I don't see any associated list of references). In the figshare file I don't see any reference to species richness values used or any indication of these actual values.

The data summary file is also not very useful. What we need is an in text file that lets the readers understand where biases and gaps in the data are.

I don't think I agree with the authors definition of spatial extent. If the unit of replication is small plots

within a large system then the relationship between disease and diversity is at the small scale not at the scale of the entire region.

Reviewer #4:

Remarks to the Author:

The question that this paper addresses is important; the paper's key contribution is that it explicitly describes some important problems in the diversity-disease literature and outlines the types of data that are needed to appropriately evaluate the biodiversity-disease relationship. However, those data do not exist at the present moment. In the absence of an appropriate dataset this paper is a "state-of-the-science" report and does not reflect how the diversity-disease relationship actually plays out in nature. Furthermore, several significant analytical errors (including inappropriate quantification of spatial scale) make me doubt the reliability of the results as they stand now. As such, I do not believe that this MS has the scientific impact that is a prerequisite for publication in Nature Communications. I list my major concerns below:

- The author's conclusions outrun the limits of their data. I would encourage the authors to restructure their manuscript to recognize that this is a meta-analysis, and that meta-analyses are limited by the available data. For example, here's a sentence from the abstract: "We provide broad evidence that biodiversity-disease relationships are generally non-linear and moderated by spatial scale; biodiversity generally inhibits disease at local scales (<100 km²) and amplifies disease at regional scales (>1,000,000 km²)." This goes too far. To appropriately recognize the limitations of the available data, I would instead write, "Our analysis reveals that biodiversity-disease relationships reported in the published literature are generally non-linear and moderated by spatial scale; among these published data, biodiversity generally inhibits disease at local scales (<100 km²) and amplifies disease at regional scales (>1,000,000 km²)." This is merely one example, but the circumspection that I suggest needs to be implemented throughout this manuscript.
- The vast majority (43 of 49 = 88%) of data points for large spatial scales (10,000+ km) come from two studies (Nguyen et al. 2016 and Wood et al. 2017). For one of those studies, each datum is essentially repeated twice (i.e., each disease has two data points, one for each time point). I realize that your random effects structure accounts for this pseudo-replication, but the reality is that your large-spatial-scale data both (1) have high leverage and (2) are extremely susceptible to inter-study variation. I realize how frustrating it is that the data needed to test your questions do not exist; I also am frustrated by this state of affairs. But the reality is that this dataset cannot do what you are trying to make it do. This is merely one example of how major gaps in the data bias these results, a problem that the authors themselves recognize: "The spatial extent of a study tended to be non-independent of study design and parasite type."
- I am confused by the use of host body size to correct for spatial extent. For the purposes of understanding the biodiversity-disease relationship, the most relevant metric is spatial scale of observation relative to spatial scale of transmission of the pathogen. Ignoring the spatial scale of transmission has already been shown to lead to spurious conclusions (e.g., Buck and Perkins 2018, Study scale determines whether wildlife loss protects against or promotes tick-borne disease). I would suggest that the authors make an estimate of each pathogen's spatial scale of transmission, and use this to adjust the spatial scale of observation rather than host body size.
- It is inappropriate to use spatial extent rather than spatial grain. Spatial grain is the unit of space to which the conclusions of a study apply. Spatial extent is the unit of space to which conclusions can be extrapolated. Luckily, spatial grain is likelier to be reported in papers than is spatial extent. Please

replace your estimates of spatial extent with estimates of spatial grain, and use spatial grain in your analyses of the influence of space on the biodiversity-disease relationship.

- It is difficult to understand from the methods section, which is very truncated, but is there any control for the appearance of particular pathogens multiple times in the dataset? The statistical models should account for the fact that some pathogens (e.g., *Batrachochytrium dendrobatidis*, WNV, *Metschnikowia bicuspidata*) are represented by multiple (sometimes many) observations.
- Given the concerns raised above regarding the appropriateness of this dataset, conclusions like the following seem extremely premature: "In general, these results suggest that biodiversity conservation can be beneficial to human health when conducted at small or intermediate scales, which are the scales at which they are most commonly implemented."
- A few mistakes with your summary of results from other papers: "As an example, in a global survey of human disease burden, disease generally increased with increasing diversity, but human schistosomiasis was negatively correlated with diversity." Here you cite Wood et al. 2017. First, I believe you mean lymphatic filariasis (just below schistosomiasis in Figure 1b of Wood et al. 2017). Second, it is disingenuous to report the negative effect for lymphatic filariasis and not the positive effect for the food-borne trematodiasis, also shown in Figure 1b of Wood et al. 2017. This kind of cherry-picking is to be avoided.
- There is a strong likelihood that the data available are biased by the choices that ecologists make about what to study, so I suggest adding a paragraph in the discussion that outlines the potential for a "file-drawer problem" in the disease ecology literature. That is, few ecologists would pursue work in a system where their data demonstrate that biodiversity promotes disease (that would make for a rather depressing career). So there is substantial potential for an overabundance of dilution effect studies relative to the frequency of dilution in nature. This should be brought out in the discussion.
- In another example of the conclusions outrunning the data, I am at a loss to understand where this overall conclusion (enunciated throughout the manuscript but here quoted from the abstract) comes from: "Despite context-dependence in biodiversity-disease relationships, biodiversity loss at local scales appears to increase disease, and thus these results suggest that biodiversity loss could have negative consequences for human and wildlife populations." I thought that you found both amplification and dilution were common? Why do you come down on the side of dilution? This does not appear to be an evidence-based conclusion.
- I realize that it is very common in ecology to use AIC for model selection, but it is also inappropriate. Your goal is to select models that fit existing data well, but AIC measures how well the model should predict new data. There are fundamental differences between models for inference and models for prediction, so it's not correct to use models optimized for prediction to do inference or vice versa. See Shmueli 2010 ("To Explain or to Predict?", *Statistical Science*) for more details.
- The writing is choppy, imprecise, and repetitive and could benefit from being professionally edited.
- I suggest using GitHub rather than Figshare for your code and data. This has several advantages: (1) GitHub makes it easier for people to find your work, including those who haven't yet read your paper but are searching for data or code associated with your questions, (2) because it doesn't require a download, people are likelier to look at your material on GitHub than on Figshare, (3) the entire history of your modifications to code and data is conveniently logged on GitHub, (4) you can continuously update material on GitHub instead of labeling files with a date and then uploading them to Figshare, which makes your workflow (5) far more reproducible.

- P5L94: "Moreover, whether hosts can dilute disease might be more observable at small scales where encounter reduction can occur, while the amplifying effect of hosts might only be observable at larger temporal and spatial scales." I understand why you might expect dilution to be more observable at small spatial scales – you've explained that well. But why should amplification be more common at larger scales? Step the reader through your rationale.
- P17L382: "Next, we quantified the monotonicity and direction of each biodiversity-disease relationship using Spearman rank correlations (see main text for methods)." I cannot find these methods.

Referees' comments to the author(s)

Referee 2:

Comments to the Authors:

In the revised version of the manuscript, the authors have sufficiently addressed my previous comments. The manuscript addresses an interesting and significant problem, and although the data set has limitations, the authors have conducted a rigorous analysis that addresses my concerns.

We wish to thank this reviewer again for their thorough and thoughtful review on the previous submission of this manuscript. We are gratified to read that our revision sufficiently addressed this reviewer's previous comments. We wish to emphasize that although the data set has limitations with respect to spatial moderation of biodiversity-disease relationships, the other key findings – that (1) biodiversity-disease relationships are commonly non-linear and (2) underrepresentation in the literature of extreme measures of diversity might result in underreporting of amplification in observational studies and dilution in experiments – are robust to characteristics of the studies and study systems. To our knowledge this analysis provides the first empirical test of these important and frequently discussed ideas.

Referee 3:

Comments to the Authors:

I find the ms much improved, but I remain unconvinced that the underlying data are sufficient and comparable enough to merit the conclusion reached (both in positive findings and in negative findings)

For instance, I am not clear how the authors analysis can control for the effects of metric on response, with these inconsistent datasets. In this case it appears the most common metric in their large scale studies (DALY) is simply not used at small scales, and is fundamentally a very different metric than all other used metrics. Similarly as far as I can tell there are NO experimental studies at large scales so I don't understand how the authors can claim evidence that there is no artifact of experiment vs. observation x spatial scale. In general what I see is lack of support for any of these factors more likely to be driven by noise, and the factors that are supported, are highly likely to be driven by biases in the underlying dataset. I think there is just a lack of sufficient comparable data to answer the questions about scale

We agree that it is important to test whether the finding regarding spatial scale was influenced by whether a study measured host richness, Shannon diversity, or Simpson's diversity, as well as whether a study measured parasite prevalence, severity (i.e., load), or disease (i.e., DALYs). In the previously submitted manuscript, we addressed this question by including a two-way interaction between spatial scale and either the diversity metric or the disease metric. Specifically, we stated that, "This result was robust to variation in how host diversity and disease were measured (Table S2). However, host richness was the only metric of host diversity that was measured across all spatial scales, and no single metric of disease was measured across spatial extents (Fig. S3), highlighting the need for more comprehensive studies of biodiversity and disease across spatial scales." To summarize this result, we found no evidence that the effect of spatial scale on biodiversity-disease relationships depended on either of these factors.

We wish to emphasize that while some disease metrics were not measured across all spatial scales, host richness and parasite prevalence were collected across more than five orders of magnitude in spatial extent. Host richness, Shannon diversity, parasite load, and parasite prevalence were all measured across nearly 10 orders of magnitude in spatial grain. We have moved the previous supplemental figures in support of these results into the main text as Figure 3. Unfortunately, it is impossible to definitively rule out whether the finding regarding spatial scale results from correlations among other variables that might change with spatial extent (such as experimental design or disease metric). However, given the analyses that we have conducted, we find little evidence in support of this hypothesis.

Despite their limitations, we feel strongly that tests of spatial scale should not be omitted from this manuscript. First, spatial scale remains a consistently hypothesized contingency in the biodiversity-disease literature, and omitting this hypothesis from our analysis would be disingenuous. Second, we were recently made aware of another manuscript that will

soon be published, which argues that dilution effects actually become stronger at larger spatial scales, and our results reveal no possible way to draw the same conclusion. Even if we drop all studies conducted at the largest spatial extent, which we now present in Appendix II, we still find that the shape of biodiversity-disease relationships are contingent on spatial scale, with significantly negative monotonic relationships (i.e., “dilution effects”) only occurring at spatial extents below 100km² and spatial grains below 100m²g⁻¹. Thus, we believe that our manuscript offers a very important counterpoint to the argument that the dilution effect becomes stronger with increasing scale.

We also wish to emphasize that tests of spatial scale only represent one of three hypotheses that this manuscript addresses, and that including tests of spatial scale in this manuscript should in no way undermine the results regarding nonlinearity and underrepresentation of the literature at the highest and lowest diversity, which are robust to potential confounders and highly novel in their own right.

Following this reviewer suggestion, we have substantially revised the section of the Results that deals with spatial moderation of the biodiversity-disease relationship. We now first discuss different metrics of spatial scale, emphasizing that the results are robust to which metric is used. We next discuss the nature and possibility of confounding between characteristics of studies and the spatial scale of biodiversity-disease relationships. Given that 5-10 orders of magnitude in spatial scale do not appear to be confounded, we conclude that across this range of scales, we can test of our hypothesis given the currently available data. We then report the results of our analysis, given these limitations, and suggest that unlike the other two hypotheses, spatial moderation of the biodiversity-disease relationship needs substantial attention in future studies. We conduct an additional analysis to test whether the results are sensitive to studies conducted at the largest spatial scale, where most confounding occurs, and while we find that some relationships are sensitive to these data, the overall pattern is robust.

Our manuscript now includes a detailed appendix (Appendix II), which details all of the analyses that we have conducted regarding spatial moderation of the biodiversity-disease relationship.

As we have stated previously, we do not think the limitations in our analysis of spatial moderation of the biodiversity-disease relationship undermines the novelty of this manuscript. We find strong support that biodiversity-disease relationships are commonly nonlinear and that underrepresentation of the literature at the highest and lowest diversity might miss important amplification and dilution effects. Furthermore, our analysis of spatial scale finds support for the well-established hypothesis that dilution effects should weaken and may even reverse in direction with increasing spatial scale. This is consistent with theory and one empirical study that conducted scale analyses on only three diseases (Cohen et al 2016 PNAS), extending these ideas to all known studies of the relationship between biodiversity and disease.

Authors conclude as the final sentence in their abstract “that biodiversity loss could have negative consequences for human and wildlife populations.” This idea recurs elsewhere in their paper Yet the authors purport to show that biodiversity loss does NOT have negative consequences (e.g. via increasing disease transmission) at large scales. Given that the scale at which biodiversity is actually declining is large (e.g. review in McGill et al 2015) and not small this seems like a major misinterpretation of their findings. Indeed, if anything their data argues the opposite – which is that at the large scale biodiversity loss is thought to be most pervasive, there are unlikely to be negative consequences for human and wildlife health.

We thank the reviewer for pointing out the McGill et al 2015 paper on biodiversity loss, and we have revised this language to be more conservative, now saying that “Despite context-dependence, biodiversity loss at local scales appears to increase disease, suggesting that at local scales, biodiversity loss could negatively impact human and wildlife populations.”

We wish to point out that there is still considerable debate surrounding the scale, location, and context of biodiversity loss (Primack et al 2018). A point which we highlight in the Introduction. Furthermore, we wish to emphasize that humans rarely encounter biodiversity change at the between-country scale where we found significant amplification, but frequently encounter biodiversity loss at the scale of a local municipality, where we found significant dilution. We therefore stand by our previous statement.

I’m afraid I can’t understand/make use of much of their data files because of missing metadata that makes it hard to understand what data was used. For instance I’m not sure what diversity “levels” are in the supp file. They do not seem to correspond with the total number of species reported in the paper cited for a couple where I checked? Although since I don’t have a reference list to correlate with the papers cited I can’t tell for certain I am cross checking correct papers (papers are numbered in figshare file but I don’t see any associated list of references). In the figshare file I don’t see any reference to species richness values used or any indication of these actual values.

We apologize for omitting useful metadata from our previous submission. We have added descriptive metadata to the supplemental data summary file. Because we assessed biodiversity-disease relationships from 218 unique datasets, it would be impractical to list every biodiversity value from each of these datasets. Therefore, in the Figshare repository, we instead provided all of the raw data that we analyzed as an r source file, along with a detailed, fully annotated R Markdown script to replicate all of the analyses that were conducted.

The data summary file, which was previously posted as a supplemental file, and which now includes descriptive metadata, includes 218 rows (one for each unique dataset) and 23 columns relating to a variety of characteristics of each study, as well as the quantitative estimates of the shape of each biodiversity-disease relationship (the dependent variables in

our analysis). This file contains the name of the dataset as it is used in the analysis as its first column, and a citation for each manuscript from which the data were collected as the second column, providing a list of every manuscript from which our data were collected.

Diversity “levels” are the number of unique diversity measurements that are included in a study (e.g., a study that compared host communities with 1, 3, and 5 species would have three diversity levels). We have revised the manuscript to clarify that we tested whether the shape of biodiversity-disease relationships depended on the number of unique values of host diversity in a study.

We are confused about the comment with respect to actual diversity values. Each of the 218 unique datasets contains at least three diversity values. It would be highly impractical to list every diversity value from every study; this is essentially the raw data that we analyzed. However, these data were made available along with the rest of our raw data in the Figshare repository. We therefore instead listed the highest and lowest diversity value for each study in the supplementary data file.

The data summary file is also not very useful. What we need is an in text file that lets the readers understand where biases and gaps in the data are.

Our description of the data includes 218 rows in its most highly summarized form (e.g., one row per dataset). An in-text table with this information would not be possible owing to space limitations on a printed sheet. Because it was not possible to summarize this quantity of information as an in-text table, we instead included this information as a data file using descriptive column titles in our submission. This file, which now includes detailed metadata, also includes a citation for each manuscript, information about each study design, hosts, and parasites, information about the spatial scale, sample size, and biodiversity measures used (including a summary of how many species were included in the most and least diverse communities for each study), and the raw variables for all of the metrics of the shape of the biodiversity-disease relationship that were used in our analyses.

To illustrate biases and gaps in the data, we previously included two multi-panel supplemental figures (Figure S3 and Figure S4), which we have now combined into a single in-text figure, Figure 3. This figure is now the first figure that we reference in the section on spatial scale, includes our estimated shape and direction of biodiversity disease relationships, and highlights the key biases and gaps in the literature, which we discuss in detail in the manuscript.

I don't think I agree with the authors definition of spatial extent. If the unit of replication is small plots within a large system then the relationship between disease and diversity is at the small scale not at the scale of the entire region.

Our results regarding spatial scale are the same regardless of whether we use spatial extent (defined by Wiens 1989 Functional Ecology as the scale over which all replicates in a study are collected) or spatial grain (defined as the scale of each individual unit of replication). We had included an analysis of spatial grain in a previous submission of this manuscript. However, reviewer comments on that submission discouraged us from including spatial grain in the manuscript, because spatial grain is more sensitive to the size of the host and scale of host-host transmission than spatial extent. We have added our previous analysis of spatial grain standardized by host biomass as a supplement in the revised manuscript (Appendix II). As we show in this supplement, our results are robust to the metric of spatial scale that we use.

We also now clarify how we define spatial scale and why we chose to report results of spatial extent in the first paragraph of the Results section titled “*Is the biodiversity-disease relationship moderated by spatial scale?*” Specifically, we now state that,

“Spatial scale can be decomposed into spatial grain, which represents the area over which a single replicate measure of biodiversity and disease are collected, and spatial extent, which represents the total area over which a study is conducted, including all measures of biodiversity and disease for a given study (Wiens 1989). Although spatial scale can be quantified in absolute terms (e.g., km²), comparing spatial scale among studies can be problematic, particularly if spatial scale is confounded with host biomass. For example, a study of bacteria could be carried out in a test-tube or in an ocean, but a study of whales could never be conducted in a test-tube. We expected this missing-cells design to be more problematic for studies of small spatial grain, which might include only a single population of a small-bodied host organism, than for studies of small spatial extents, which must always include multiple replicate host communities, by design. We therefore report the results of unstandardized spatial extent, noting that the results were qualitatively similar for spatial extent standardized by host biomass and for spatial grain standardized by host biomass (Appendix II).”

Referee 4:

Comments to the Authors:

The question that this paper addresses is important; the paper's key contribution is that it explicitly describes some important problems in the diversity-disease literature and outlines the types of data that are needed to appropriately evaluate the biodiversity-disease relationship. However, those data do not exist at the present moment. In the absence of an appropriate dataset this paper is a "state-of-the-science" report and does not reflect how the diversity-disease relationship actually plays out in nature. Furthermore, several significant analytical errors (including inappropriate quantification of spatial scale) make me doubt the reliability of the results as they stand now. As such, I do not believe that this MS has the scientific impact that is a prerequisite for publication in Nature Communications. I list my major concerns below:

First, as described in our response to Reviewer 2, we wish to emphasize that we have sufficiently well-replicated data sets to test the questions that we address in this study. This study presented the results of more than 200 biodiversity-disease relationships on 67 parasite species. These data are unquestionably sufficient to address whether biodiversity-disease relationships are generally non-monotonic and whether they are unskewed or right or left skewed, extremely important and unanswered questions. Moreover, these data are sufficient to assess whether underrepresentation in the literature of extreme measures of diversity might result in underreporting of amplification in observational studies and dilution in experiments. For our third question about scale, these data are sufficient to assess the effect of scale in an unconfounded manner across over 5 orders of magnitude of the scale gradient (from less than 0.1km² to more than 100,000 km²). The only scale at which things become confounded is at 1,000,000 km², because all the studies at this scale are observational and half of the relationships use DALYs as their endpoint. So, we discourage throwing the baby out with the bathwater. The data have limitations that we highlight in our paper, but they still allow us to make important scientific advances to the biodiversity-disease literature. To argue that the breadth of studies in this database somehow "does not reflect how diversity-disease relationships actually play out in nature" appears surprisingly disingenuous given that many of the >200 studies are field observational studies and many of the experimental studies are not obviously contrived.

Furthermore, the suggested analytical errors described below are inconsequential, and do not qualitatively change our results nor their interpretation.

The author's conclusions outrun the limits of their data. I would encourage the authors to restructure their manuscript to recognize that this is a meta-analysis, and that meta-analyses are limited by the available data. For example, here's a sentence from the abstract: "We provide broad evidence that biodiversity-disease relationships are generally non-linear and moderated by spatial scale; biodiversity generally inhibits disease at local scales (<100 km²) and amplifies disease at regional scales (>1,000,000 km²)." This goes too far. To appropriately recognize the

limitations of the available data, I would instead write, “Our analysis reveals that biodiversity-disease relationships reported in the published literature are generally non-linear and moderated by spatial scale; among these published data, biodiversity generally inhibits disease at local scales (<100 km²) and amplifies disease at regional scales (>1,000,000 km²).” This is merely one example, but the circumspection that I suggest needs to be implemented throughout this manuscript.

We now emphasize that the meta-analysis is limited by the available data and have made the recommended revision. Thank you for this suggestion.

The vast majority (43 of 49 = 88%) of data points for large spatial scales (10,000+ km) come from two studies (Nguyen et al. 2016 and Wood et al. 2017). For one of those studies, each datum is essentially repeated twice (i.e., each disease has two data points, one for each time point). I realize that your random effects structure accounts for this pseudo-replication, but the reality is that your large-spatial-scale data both (1) have high leverage and (2) are extremely susceptible to inter-study variation. I realize how frustrating it is that the data needed to test your questions do not exist; I also am frustrated by this state of affairs. But the reality is that this dataset cannot do what you are trying to make it do. This is merely one example of how major gaps in the data bias these results, a problem that the authors themselves recognize: “The spatial extent of a study tended to be non-independent of study design and parasite type.”

We thank the reviewer for highlighting the Nguyen et al. 2016 study in addition to the Wood et al. 2017 study. Thanks to this reviewer comment, we realized that we erroneously coded two studies (Nguyen et al. 2016 and Derne et al. 2011) as occurring at smaller spatial extents than the actual extent of those studies (which was at the same spatial extent as the Wood et al. 2017 study). Correcting this error does not change the results of our analyses qualitatively, and gives us further confidence in the robustness of our results. In response to this comment from the reviewer, we also performed a sensitivity analysis on our test of the effect of spatial scale on the shape of biodiversity-disease relationships, testing whether the effect of scale on shape was robust to inclusion of the Wood et al. 2017 study, as well as all three studies that were conducted at the largest spatial extent. We provide the results of this sensitivity analysis in the supplemental materials, Appendix II. Removing these studies does alter our ability to detect the effect of spatial-scale on the biodiversity-disease relationship in some models. Specifically, spatial scale becomes non-significant in the models that test Spearman’s Rho when we exclude either the Wood et al 2017 study or all three studies that were conducted at the largest spatial extent. However, we are still able to recover a significant relationship in the model that tests Pearson’s skewness, regardless of whether the Wood et al 2017 study or all three studies at the largest extents are omitted from our analysis. Furthermore, when all three studies conducted at the largest spatial extent are excluded from the analysis, we still recover a marginally significant ($p=0.055$) relationship between spatial grain and Pearson’s skewness. These results are detailed in Appendix II. Importantly, even when we exclude studies at scales where we observed significant positive monotonic and left-skewed relationships (indicative of amplification effects), we must still reject the hypotheses that dilution is spatially invariant or becomes

stronger at larger spatial scales. In other words, we consistently find evidence that the occurrence of dilution effects depends on spatial scale, and that even in models where we exclude datasets with high leverage, we still find evidence in support of “dilution effects” (e.g., significantly negative monotonic and right skewed relationships) only at spatial extents below 100km² and at spatial grains below 100m²g⁻¹.

Nevertheless, we now lead our discussion of the spatial analysis with a strong emphasis on the caveats associated with the data before proceeding with the results.

I am confused by the use of host body size to correct for spatial extent. For the purposes of understanding the biodiversity-disease relationship, the most relevant metric is spatial scale of observation relative to spatial scale of transmission of the pathogen. Ignoring the spatial scale of transmission has already been shown to lead to spurious conclusions (e.g., Buck and Perkins 2018, Study scale determines whether wildlife loss protects against or promotes tick-borne disease). I would suggest that the authors make an estimate of each pathogen’s spatial scale of transmission, and use this to adjust the spatial scale of observation rather than host body size.

We used host body size to correct for spatial extent in response to a previous reviewer’s suggestion that the effect of spatial scale might be dependent on the scale of host-host transmission. Because the scale of host-host transmission is unreported or unknown for most pathogens, we instead used host biomass. We recognize that the scale of transmission would be an ideal standardizing variable, but these data are unavailable or unknown for most parasites. We highlight this lack of information in the manuscript, stating that “Furthermore, the scale associated with data used does not necessarily mean that this is the scale of the transmission cycle. It is therefore possible that some studies conducted at small spatial scales did not capture the entire parasite transmission cycle compared to studies conducted at large spatial scales”. The important point to understand is that we have conducted the analyses with several metrics of scale and get qualitatively the same results, highlighting that the results are robust to the measure of scale.

It is inappropriate to use spatial extent rather than spatial grain. Spatial grain is the unit of space to which the conclusions of a study apply. Spatial extent is the unit of space to which conclusions can be extrapolated. Luckily, spatial grain is likelier to be reported in papers than is spatial extent. Please replace your estimates of spatial extent with estimates of spatial grain, and use spatial grain in your analyses of the influence of space on the biodiversity-disease relationship.

We included spatial grain (with qualitatively similar results) in a previously submitted version of this manuscript. However, we ultimately omitted spatial grain from the manuscript because spatial grain is likely to be much more sensitive to host biomass and/or the scale of host-host transmission than spatial extent.

We now include an additional analysis of spatial grain (standardized host biomass) as Appendix II. The analysis of spatial grain is consistent with the analysis of spatial extent,

showing dilution effects more commonly occurring at small spatial scales and amplification effects more commonly occurring at large spatial scales.

We additionally added a new paragraph to the Results section, titled “Is the biodiversity-disease relationship moderated by spatial scale?” that better explains our use of spatial extent. Specifically, we now state that,

“Spatial scale can be decomposed into spatial grain, which represents the area over which a single replicate measure of biodiversity and disease are collected, and spatial extent, which represents the total area over which a study is conducted, including all measures of biodiversity and disease for a given study (Wiens 1989). Although spatial scale can be quantified in absolute terms (e.g., km²), comparing spatial scale among studies can be problematic, particularly if spatial scale is confounded with host biomass. For example, a study of bacteria could be carried out in a test-tube or in an ocean, but a study of whales could never be conducted in a test-tube. We expected this missing-cells design to be more problematic for studies of small spatial grain, which might include a single population of a small-bodied host organism, than for studies of small spatial extents, which must always include multiple replicate host communities, by design. We therefore report the results of unstandardized spatial extent, noting that the results were qualitatively similar for spatial extent standardized by host biomass and for spatial grain standardized by host biomass (Appendix II).”

It is difficult to understand from the methods section, which is very truncated, but is there any control for the appearance of particular pathogens multiple times in the dataset? The statistical models should account for the fact that some pathogens (e.g., *Batrachochytrium dendrobatidis*, WNV, *Metschnikowia bicuspidata*) are represented by multiple (sometimes many) observations.

We thank the reviewer for this suggestion. We now include pathogen as a random effect in our model structure. Including pathogen as a random effect in the model does not alter the results, which is emphasized in the manuscript and the results from the model are provided in the Supplement.

Given the concerns raised above regarding the appropriateness of this dataset, conclusions like the following seem extremely premature: “In general, these results suggest that biodiversity conservation can be beneficial to human health when conducted at small or intermediate scales, which are the scales at which they are most commonly implemented.”

As noted in our response to Reviewer 3, we have toned down the language regarding conservation implications for our results.

A few mistakes with your summary of results from other papers: “As an example, in a global survey of human disease burden, disease generally increased with increasing diversity, but human schistosomiasis was negatively correlated with diversity.” Here you cite Wood et al. 2017. First, I believe you mean lymphatic filariasis (just below schistosomiasis in Figure 1b of

Wood et al. 2017). Second, it is disingenuous to report the negative effect for lymphatic filariasis and not the positive effect for the food-borne trematodiasis, also shown in Figure 1b of Wood et al. 2017. This kind of cherry-picking is to be avoided.

The purpose of this sentence is simply to highlight that amplification is not universal at large scales and dilution is not universal at small scales. We have removed this description of the Wood et al 2017 study from the revised manuscript and replaced it with a description of the study by Derne et al 2011. Specifically, this statement now reads, “in a global survey of island nations, leptospirosis incidence strongly declined with increasing mammal species richness.”

There is a strong likelihood that the data available are biased by the choices that ecologists make about what to study, so I suggest adding a paragraph in the discussion that outlines the potential for a “file-drawer problem” in the disease ecology literature. That is, few ecologists would pursue work in a system where their data demonstrate that biodiversity promotes disease (that would make for a rather depressing career). So there is substantial potential for an overabundance of dilution effect studies relative to the frequency of dilution in nature. This should be brought out in the discussion.

We have added a paragraph discussing the potential file drawer problem as we understand it. However, we wish to note that a previous study conducted in 2015 found no evidence of a systematic bias against reporting amplification effects for parasites of humans (e.g., Civitello et al 2015).

The paragraph states, “The results of this analysis are limited by the availability of previously published data, which may be biased to report significant effects (i.e., a “file-drawer problem”) or to select systems to study that are likely to show dilution (i.e., to show “system selection bias”; Wood & Lafferty 2013, Salkeld et al 2015). However, a previous study conducted in 2015 found no evidence of a file drawer problem for parasites of humans (Civitello et al 2015). Furthermore, by re-analyzing previously published data, our study is somewhat less sensitive to the file-drawer problem than typical meta-analyses, which rely on previously published statistical tests. Nevertheless, we cannot rule out the possibility for an overabundance of studies finding significant dilution or amplification relative to the frequency of dilution or amplification in nature”

In another example of the conclusions outrunning the data, I am at a loss to understand where this overall conclusion (enunciated throughout the manuscript but here quoted from the abstract) comes from: “Despite context-dependence in biodiversity-disease relationships, biodiversity loss at local scales appears to increase disease, and thus these results suggest that biodiversity loss could have negative consequences for human and wildlife populations.” I thought that you found both amplification and dilution were common? Why do you come down on the side of dilution? This does not appear to be an evidence-based conclusion.

There is a preponderance of evidence (both theoretical, and empirical) that dilution effects occur at small scales, and our analysis adds further support to this hypothesis, regardless of the relationship between spatial scale and the biodiversity-disease relationship. In contrast, we found significant amplification only at the largest spatial grains and extent, and this result was strongly influenced by a single study on biodiversity-disease relationships that used a distinct metric of disease, making this result much less clear. Regardless of whether there is amplification at large scales, and as described in our response to Reviewer 3, humans and wildlife most commonly encounter biodiversity-loss at small spatial scales, thus biodiversity loss could have negative consequences for human and wildlife populations.

Nevertheless, we have edited this statement to be more conservative, now stating, “Despite context-dependence, biodiversity loss at local scales appears to increase disease, suggesting that at local scales, biodiversity loss could negatively impact human and wildlife populations.”

I realize that it is very common in ecology to use AIC for model selection, but it is also inappropriate. Your goal is to select models that fit existing data well, but AIC measures how well the model should predict new data. There are fundamental differences between models for inference and models for prediction, so it's not correct to use models optimized for prediction to do inference or vice versa. See Shmueli 2010 (“To Explain or to Predict?”, *Statistical Science*) for more details.

Following Shmueli 2010, we repeated our model selection procedure using BIC instead of AIC. The model-estimated effects were qualitatively similar using the more conservative, BIC for model selection. Specifically, although this change in the metric used for model-comparison reduced the number of significant biodiversity-disease relationships that were observed as well as the number of non-linear biodiversity-disease relationships overall, it did not qualitatively change the result that biodiversity-disease relationships were commonly nonlinear as expected. 49% of studies were best fit by linear, second-order, or third-order polynomial; of these, 53% exhibited non-linear relationships. In the Supplemental materials, we now include a table of results including both AIC and BIC. We also state in the Results that, “These effects were qualitatively similar using the more conservative, BIC for model selection (49% of studies were best fit by linear, second-order, or third-order polynomial; of these, 53% exhibited non-linear relationships; Table S1)”.

We also wish to emphasize that this analysis is only one of three analyses that are presented in the manuscript confirming that the shape of the biodiversity-disease relationship is nonlinear. Using BIC for model selection in this analysis has no impact on our measure of the monotonicity and direction of biodiversity disease relationships (measured using Spearman's Rho), or the skewness of biodiversity-disease relationships (measured using Pearson's skewness).

The writing is choppy, imprecise, and repetitive and could benefit from being professionally edited.

We are disappointed to read that this reviewer found our writing unsatisfactory. No other reviewers raised concerns about the writing quality. If there are particular locations where the reviewer feels that editing is needed, please let us know. Otherwise, the senior author could very well be considered a professional editor at this point in his career (>150 peer-reviewed publications, he has published >13 manuscripts this year already, and is a professional editor for several top-tier disease and ecology journals), and there will also be professional copy editing on this manuscript after it is accepted for publication.

I suggest using GitHub rather than Figshare for your code and data. This has several advantages: (1) GitHub makes it easier for people to find your work, including those who haven't yet read your paper but are searching for data or code associated with your questions, (2) because it doesn't require a download, people are likelier to look at your material on GitHub than on Figshare, (3) the entire history of your modifications to code and data is conveniently logged on GitHub, (4) you can continuously update material on GitHub instead of labeling files with a date and then uploading them to Figshare, which makes your workflow (5) far more reproducible.

The first author of this paper is not fluent with GitHub and we have chosen instead to publish all data and code necessary to replicate the analyses on the public repository, Figshare. Moreover, we have published an RMarkdown file, which we believe provides an extremely high level of transparency and reproducibility. We will consider learning how to use GitHub for future studies.

P5L94: “Moreover, whether hosts can dilute disease might be more observable at small scales where encounter reduction can occur, while the amplifying effect of hosts might only be observable at larger temporal and spatial scales.” I understand why you might expect dilution to be more observable at small spatial scales – you’ve explained that well. But why should amplification be more common at larger scales? Step the reader through your rationale.

We thank the reviewer for this comment and now gladly step the reader through our rationale. Specifically, we now state in the Introduction that, “Context dependence in the biodiversity-disease relationship may also arise when the direction of the biodiversity-disease relationship depends on the spatial scale of observation^{2,8,20}. Local processes influence the abundance of species at relatively small spatial scales, while regional processes influence the distributions of species across large spatial extents²¹. Relying on this well-characterized ecological phenomenon, it has been proposed that biodiversity-disease relationships should be strongest at local scales, where biotic interactions are most likely to occur, and should weaken or could even reverse at larger scales, where individual studies encompass a greater diversity of habitat types and abiotic factors like climate may cause the distributions of hosts and parasites to covary^{18,22}. In other words, at small spatial

scales, increasing biodiversity might cause a reduction in parasite abundance, resulting in an observed dilution effect, while at larger spatial scales, regional processes might cause host biodiversity and disease risk to positively covary, offsetting the dilution effect or even resulting in an observed amplification effect. Moreover, whether hosts can dilute disease might be more observable at small scales where encounter reduction can occur, while the amplifying effect of hosts might only be observable at larger temporal and spatial scales
23.”

P17L382: “Next, we quantified the monotonicity and direction of each biodiversity-disease relationship using Spearman rank correlations (see main text for methods).” I cannot find these methods.

These methods were previously described at Line 139 of the manuscript. Specifically, we stated that, “we used Spearman rank correlation tests (not constrained to pass through the origin), which make no assumption about the underlying distribution of the data nor the linearity of the relationship between variables, and are therefore not constrained by the functional form of the biodiversity-disease relationship. We quantified whether each biodiversity-disease relationship was monotonic and positive (disease increases, but may level off, as diversity increases), monotonic and negative (disease decreases but may level off as diversity increases), or non-monotonic (disease increases with diversity at low levels, but eventually decreases at high enough diversity; Fig. 1a). The estimated Spearman rank correlation coefficient (Rho) approaches one for monotonic, positive relationships, and approaches negative one for monotonic, negative relationships. We therefore used Rho to define monotonic amplification ($Rho > 0, p < 0.05$), monotonic dilution ($Rho < 0, p < 0.05$), and non-significant or non-monotonic relationships ($p > 0.05$).”.

We have edited the text in the methods to make this text easier to find, so that the text now reads, “Next we quantified the monotonicity and direction of each biodiversity-disease relationship using Spearman rank correlations (see the Results subsection titled “What is the shape of the biodiversity-disease relationship?” for methods). Briefly, the Spearman rank correlation coefficient (Rho) and its associated p-value were used to define monotonic amplification ($Rho > 0, p < 0.05$), monotonic dilution ($Rho < 0, p < 0.05$), and non-significant or non-monotonic relationships ($p > 0.05$).”.

Reviewers' Comments:

Reviewer #4:

Remarks to the Author:

The authors have now addressed some of my concerns about their analysis, although my worries remain regarding the fidelity with which their dataset (i.e., the existing literature) reflects the biodiversity–disease relationship in nature. Take for example the following comment:

“To argue that the breadth of studies in this database somehow “does not reflect how diversity-disease relationships actually play out in nature” appears surprisingly disingenuous given that many of the >200 studies are field observational studies and many of the experimental studies are not obviously contrived.”

I made this comment with reference to the “system selection bias” problem that the authors now discuss in the text of their MS – so how is this comment disingenuous if the authors agree with it?

I appreciate that the authors went to great lengths to perform a sensitivity analysis on the test of the effect of spatial scale on the shape of the biodiversity-disease relationship, to show that results are similar whether spatial grain or spatial extent is used, to include pathogen identity as a random effect in models where appropriate, and to temper their language and constrain their conclusions.

The authors missed the point of the Shmueli et al. study – there should be a measure of goodness-of-fit used for model selection (e.g., R^2), not a measure of predictive power (e.g., AIC or BIC).

There needs to be more discussion on the use of host body size in the text of the manuscript. You’re using it as a proxy for the scale of transmission, so please explain that.

Referees' comments to the author(s)

Referee 4:

Comments to the Authors:

The authors have now addressed some of my concerns about their analysis, although my worries remain regarding the fidelity with which their dataset (i.e., the existing literature) reflects the biodiversity–disease relationship in nature. Take for example the following comment:

“To argue that the breadth of studies in this database somehow “does not reflect how diversity-disease relationships actually play out in nature” appears surprisingly disingenuous given that many of the >200 studies are field observational studies and many of the experimental studies are not obviously contrived.”

I made this comment with reference to the “system selection bias” problem that the authors now discuss in the text of their MS – so how is this comment disingenuous if the authors agree with it?

We thank the reviewer for clarifying their position regarding system selection bias. We agree that we cannot rule out the possibility for an overabundance of studies finding significant dilution or amplification relative to the frequency of dilution or amplification in nature, and we therefore included this text in the manuscript. However, saying that we cannot rule out the possibility of system selection bias is not the same thing as saying that there *is* a system selection bias. We have neither evidence for nor against system selection bias in these data; yet, we *do* know that many of the >200 studies in this manuscript are field observational studies and that many of the experimental studies are not obviously contrived. Many of these studies also report non-significant biodiversity-disease relationships, which would be inconsistent with a system selection bias in the literature. We are willing mention system selection bias as a possibility for which we have no evidence, but this possibility in no way suggests "that the breadth of studies in this database somehow “does not reflect how diversity-disease relationships actually play out in nature".

I appreciate that the authors went to great lengths to perform a sensitivity analysis on the test of the effect of spatial scale on the shape of the biodiversity-disease relationship, to show that results are similar whether spatial grain or spatial extent is used, to include pathogen identity as a random effect in models where appropriate, and to temper their language and constrain their conclusions.

We wish to reiterate our gratitude for these suggestions by the reviewer. We believe that these suggestions improved the manuscript considerably.

The authors missed the point of the Shmueli et al. study – there should be a measure of goodness-of-fit used for model selection (e.g., R^2), not a measure of predictive power (e.g.,

AIC or BIC).

We are grateful to the reviewer for clarifying their request regarding model selection. We have now replicated our model selection analysis using the adjusted R^2 from each model. This model selection analysis resulted in qualitatively similar results to AIC and BIC: Out of the 205 studies that included more than three levels of biodiversity, 80% were best fit by a linear, second-order, or third-order polynomial model (i.e., exhibited a relationship between biodiversity and disease). Of these studies, biodiversity-disease relationships were still most commonly non-linear, as predicted. More specifically, 71% exhibited non-linear relationships (either second- or third-order polynomial), while 5% exhibited a linear, positive biodiversity-disease relationship (e.g., linear amplification effect), and 24% exhibited a linear, negative biodiversity-disease relationship (e.g., linear dilution effect).

We had previously interpreted the following paragraph (particularly the last sentence) from Shmueli 2010 (“To Explain or to Predict?”, *Statistical Science*) as suggesting that BIC should be used for explanatory model selection, “As mentioned in Section 1.6, the statistics literature on model selection includes a rich discussion on the difference between finding the “true” model and finding the best predictive model, and on criteria for explanatory model selection versus predictive model selection. A popular predictive metric is the in-sample Akaike Information Criterion (AIC). Akaike derived the AIC from a predictive viewpoint, where the model is not intended to accurately infer the “true distribution,” but rather to predict future data as accurately as possible (see, e.g., Berk, 2008; Konishi and Kitagawa, 2007). Some researchers distinguish between AIC and the Bayesian information criterion (BIC) on this ground. Sober (2002) concluded that AIC measures predictive accuracy while BIC measures goodness of fit”.

The revised manuscript now includes model selection using AIC in the main text with BIC and adjusted R^2 as selection criteria in the supplemental materials. We also present the adjusted R^2 of each model in Table S1, and provide a supplemental figure showing the adjusted R^2 of the best model for each model selection approach. As can be seen from this new supplemental figure, we are not selecting the best of uniformly bad models. Our models do account for a considerable amount of variation.

There needs to be more discussion on the use of host body size in the text of the manuscript. You’re using it as a proxy for the scale of transmission, so please explain that.

We have updated the manuscript to provide a more in-depth discussion of host body size in our analyses. We included host body mass in our analyses for two reasons.

First, we included host body mass because comparisons of different spatial scales can be confounded by host biomass. We explain this rationale in the Results section titled, “Is the biodiversity-disease relationship moderated by spatial scale?” Specifically, we state that, “Although spatial scale can be quantified in absolute terms (e.g., km^2), comparing spatial scale among studies can be problematic, particularly if spatial scale is confounded with host biomass. For example, a study of bacteria could be carried out in a test-tube or in an ocean, but a study of whales could never be conducted in a test-tube. We expected this missing-cells design to be more problematic for studies of small spatial grain, which might

include a single population of a small-bodied host organism, than for studies of small spatial extents, which must always include multiple replicate host communities, by design.”

Second, we included host body mass to account for the fact that the effect of spatial scale might be dependent on the scale of host-host transmission. We now explain this rationale more thoroughly in the manuscript stating that, “Furthermore, the scale associated with data used does not necessarily mean that this is the scale of the transmission cycle, and ignoring the spatial scale of transmission can lead to spurious conclusions. For example, depending on the scale of transmission, study scale can determine whether wildlife loss protects against or promotes tick-borne disease (Buck and Perkins 2018). It is therefore possible that some studies conducted at small spatial scales did not capture the entire parasite transmission cycle compared to studies conducted at large spatial scales. Ideally, an analysis of spatial scale would therefore include transmission scale as a standardizing variable. However, because the scale of transmission is unreported or unknown for most pathogens, we instead used host biomass, because transmission often scales with host biomass (Han et al. 2015, Bordes et al. 2009). Our results were robust to standardization by host biomass (Fig. S5).”

Reviewers' Comments:

Reviewer #4:

Remarks to the Author:

The authors have now addressed my concerns about explanation versus prediction (i.e., R^2 versus BIC versus AIC) and host body size. However, concerns remain for me about whether the actual dataset (i.e., values extracted from papers) reflects nature. I disagree with the authors when they say that we have "neither evidence for nor against system selection bias in these data." Consider Wood and Lafferty 2013 (Biodiversity and disease: a synthesis of ecological perspectives on Lyme disease transmission, TREE), who show that, while studies in the ecological literature tend to find dilution of Lyme disease, similar studies in the epidemiological literature tend to find amplification. How can researchers working on the same system come to opposite conclusions? Because the way that studies are framed influences their outcome, the perspective of researchers influences the outcome. Yes, many of the studies in the dataset are "observational" or experimental studies that are "not obviously contrived" - but humans are choosing where and when to do these studies, and humans are influenced by their outlook - hence, observational studies from ecologists tell a different story than observational studies from epidemiologists. I'm gratified that the authors recognize this tangentially, but I think that there needs to be more circumspection throughout the manuscript - the results reflect the state of the literature, not necessarily the state of nature.

Referees' comments to the author(s)

Referee 4:

Comments to the Authors:

The authors have now addressed my concerns about explanation versus prediction (i.e., R2 versus BIC versus AIC) and host body size. However, concerns remain for me about whether the actual dataset (i.e., values extracted from papers) reflects nature. I disagree with the authors when they say that we have "neither evidence for nor against system selection bias in these data." Consider Wood and Lafferty 2013 (Biodiversity and disease: a synthesis of ecological perspectives on Lyme disease transmission, TREE), who show that, while studies in the ecological literature tend to find dilution of Lyme disease, similar studies in the epidemiological literature tend to find amplification. How can researchers working on the same system come to opposite conclusions? Because the way that studies are framed influences their outcome, the perspective of researchers influences the outcome. Yes, many of the studies in the dataset are "observational" or experimental studies that are "not obviously contrived" - but humans are choosing where and when to do these studies, and humans are influenced by their outlook - hence, observational studies from ecologists tell a different story than observational studies from epidemiologists. I'm gratified that the authors recognize this tangentially, but I think that there needs to be more circumspection throughout the manuscript - the results reflect the state of the literature, not necessarily the state of nature.

Response: We thank the reviewer for clarifying their position regarding whether the published literature sufficiently represents nature, and have added additional circumspection throughout the manuscript to address this concern. Below, we specify the locations in the manuscript where we note that the results reflect the state of the literature, not necessarily the state of nature:

- (1) We added explicit reference to this issue in the paragraph describing limitations of our analysis of spatial scale. Specifically, we now state that, "...observational studies and studies of human pathogens both tended to occur at larger spatial scales than manipulative studies and studies of wildlife pathogens (Fig. 3). This collinearity highlights an important limitation in the study of biodiversity-disease relationships: our understanding of the relationship between biodiversity and disease is limited by research priorities, approaches, and study systems, which can vary among individual research groups (Wood & Lafferty 2013, Johnson et al. 2015). Consequently, we cannot rule out the possibility that these results could change if future studies filled these research gaps, allowing tests of these context dependencies to be less collinear."
- (2) In the abstract we begin our description of the results by stating that "Our analysis of the published literature reveals..."
- (3) In the introduction, we note that our understanding of spatial moderation of biodiversity-disease relationships is limited because, "few studies have been conducted across multiple spatial scales."

- (4) We edited the statement describing the study in the introduction to highlight our focus on published literature, so that it now says, “Specifically, we test whether previously published biodiversity-disease relationships are generally (a) non-linear, (b) moderated by spatial scale, and (c) sensitive to underrepresentation in the literature of extremely low and high diversity. Our results indicate that among published data, biodiversity-disease relationships are generally non-linear...”
- (5) We begin the Results and Discussion section by stating, “First, we tested whether the published relationship between biodiversity and disease was...”
- (6) In the same paragraph, we state, “Importantly, these models were only based on the data presented in each study...”
- (7) In the section of the results on the effects of spatial scale, we now state that, “In the published literature, spatial extent tended to be correlated with the metrics used to estimate diversity and disease...”
- (8) We also now state that, “These results reveal an association between the shape of published biodiversity-disease relationships and the spatial scale of observations, supporting the hypothesis that biodiversity-disease relationships are scale-dependent”
- (9) To better highlight the current state of the literature, we changed the title of the final section of Results and Discussion to, “*Underrepresentation in the literature*”, and begin this section by stating, “Finally, we tested the hypothesis that underrepresentation in the literature of extreme environmental scenarios with either high or low biodiversity communities in experimental and observational studies might bias studies to more commonly report amplification and dilution effects, respectively.”
- (10) We edited a statement in this subsection so that it now reads, “These results indicate that scale-dependence of published biodiversity-disease relationships is robust to underrepresentation of communities at low diversity levels.”
- (11) We edited a statement that previously read, “dilution effects most commonly occur...” So that it now reads, “dilution effects are most commonly observed...”
- (12) Finally, we edited the penultimate sentence of the manuscript so that it now reads, “This study provides quantitative evidence that, among published studies, the relationship between biodiversity and disease is non-linear and scale-dependent”